# False Coverage Proportion Control for Conformal Prediction

**Alexandre Blain** [1 2 3]  **Bertrand Thirion** [1 4 2]  **Pierre Neuvial** [3]

## Abstract

Split Conformal Prediction (SCP) provides a computationally efficient way to construct confidence intervals in prediction problems. Notably, most of the theory built around SCP is focused on the single test point setting. In real-life, inference sets consist of multiple points, which raises the question of coverage guarantees for many points simultaneously. While *on average*, the False Coverage Proportion (FCP) remains controlled, it can fluctuate strongly around its mean, the False Coverage Rate (FCR). We observe that when a dataset is split multiple times, classical SCP may not control the FCP in a majority of the splits. We propose CoJER, a novel method that achieves sharp FCP control in probability for conformal prediction, based on a recent characterization of the distribution of conformal $p$-values in a transductive setting. This procedure incorporates an aggregation scheme which provides robustness with respect to modeling choices. We show through extensive real data experiments that CoJER provides FCP control while standard SCP does not. Furthermore, CoJER yields shorter intervals than the *state-of-the-art* method for FCP control and only slightly larger intervals than standard SCP.

## 1. Introduction

In the realm of uncertain predictions, conformal prediction methods provide a robust approach to obtaining confidence intervals that maintain a predetermined coverage with statistical guarantees. Introduced by Vovk et al., 2005, conformal prediction methods are nonparametric techniques that can be applied to any predictive model to generate prediction intervals that are valid under minimal assumptions about the underlying data distribution. These methods are used across various fields such as medicine, finance, and engineering, where decision-making requires quantifiable estimates of uncertainty.

Split conformal prediction (SCP) is the standard practical approach within the conformal prediction framework, offering improved computational efficiency and scalability. In this method, the dataset is divided into two parts: a training set and a calibration set. The model is first trained on the training set, and then the calibration set is used to compute non-conformity scores and ultimately confidence intervals. This approach simplifies the computation compared to full conformal prediction, as the calibration set remains fixed while the model parameters are tuned. Split conformal prediction ensures marginal coverage, meaning that on average, the prediction intervals will contain the true value a specified proportion of the time – e.g., $90\%$ or $95\%$ – providing a reliable measure of uncertainty.

Many real-world applications, such as healthcare (predicting outcomes for multiple patients), finance (forecasting ranges for multiple assets), and manufacturing (quality control across batches), inherently require predictions for multiple data points simultaneously. However, SCP encounters limitations when applied to multiple test points simultaneously, as discussed by Vovk, 2013. A key concern in this transductive setting is to control the false coverage proportion (FCP), that is, the proportion of prediction intervals that fail to contain the true value. In applications involving a large number of simultaneous predictions, it is important to ensure that the FCP remains within acceptable bounds. While SCP controls the *expected* proportion of non-covered points (FCR; False Coverage Rate), it does not guarantee control over the actual proportion of non-covered intervals across multiple test points with high probability.

To address this, we introduce a method to control the FCP using conformal $p$-values within a Joint Error Rate (JER, Blanchard et al., 2020) control framework and cumulative distribution function (CDF) formulations (Gazin et al., 2024). This approach allows controlling the FCP with high probability across multiple predictions, ensuring that the proportion of incorrect intervals does not exceed a pre-specified level. While the approach of Gazin et al., 2024 yields valid FCP bounds, the latter can be conservative, as

[1] INRIA [2] Université Paris-Saclay [3] Institut de Mathématiques de Toulouse, Université de Toulouse; CNRS; UPS, F-31062 Toulouse Cedex 9, France [4] CEA. Correspondence to: Alexandre Blain <alexandre.blain@inria.fr>.

*Proceedings of the $42^{nd}$ International Conference on Machine Learning*, Vancouver, Canada. PMLR 267, 2025. Copyright 2025 by the author(s).

discussed by the authors. We propose a novel method that relies on nonparametric calibration to obtain sharper FCP bounds.

Furthermore, to mitigate the impact of modeling decisions, we extend this procedure to aggregate conformal prediction intervals given by different models. Notably, we manage to obtain a fully nonparametric method *valid for any aggregation scheme*, which ensures valid coverage while maintaining relatively small prediction intervals.

Through extensive experimental validation on OpenML datasets, we demonstrate that our proposed methods effectively control the FCP and produce sharp prediction intervals. We also show that the proposed aggregation procedure retains FCP control while yielding more informative intervals than existing methods. Our results highlight the practical utility of these contributions in improving the reliability and interpretability of conformal prediction methods.

## 2. Refresher on Conformal Prediction

**Notation.** The input data are denoted by $(X_1, Y_1), \ldots, (X_n, Y_n)$ with $(X_i, Y_i) \in \mathbb{R}^p \times \mathbb{R}$, where $n$ is the number of samples and $p$ the number of variables. For any set $S$, $|S|$ denotes the cardinality of $S$. For any positive integer $k$, $[\![k]\!]$ denotes the set $\{1, \ldots, k\}$. For a vector $\mathbf{z} = (z_j)_{j \in [\![p]\!]}$ and $S \subset [\![p]\!]$, we denote by $z_{(j:S)}$ (or $z_{(j)}$ when there is no ambiguity) the $j^{th}$ smallest value in the sub-vector $(z_s)_{s \in S}$. Equality in distribution is denoted by $\stackrel{d}{=}$, $\mathfrak{S}_n$ is the symmetric group of degree $n$, and $\mathcal{U}[0, 1]$ denotes the uniform distribution on $[0, 1]$.

**Problem setup.** The goal is to build a confidence interval given a predictive model and a new sample $X_{n+1} \in \mathbb{R}^p$. The associated true outcome $Y_{n+1}$ is not observed but should lie inside the confidence interval with high probability. Formally, the goal is to build a confidence interval $\widehat{C}_\alpha$ such that:

$$\mathbb{P}\left\{Y_{n+1} \in \widehat{C}_\alpha\left(X_{n+1}\right)\right\} \geq 1 - \alpha. \qquad (1)$$

### 2.1. Split Conformal Prediction

Split Conformal Prediction (SCP) offers a simple and flexible approach to constructing reliable prediction intervals. This method offers the desired guarantee in a model-agnostic way at the cost of splitting the $n$ observations into a training set $\mathcal{D}_{train}$ and a calibration set $\mathcal{D}_{cal}$. For notation simplicity, we assume that the calibration set $\mathcal{D}_{cal}$ consists of all $n$ observations, and that an external training data set $\mathcal{D}_{train}$ (say $(X_i, Y_i)_{-t+1 \leq i \leq 0}$) is also available.

The intuition behind SCP is that the calibration set provides

a realistic measure of the performance of the trained model. We consider arbitrary non-conformity scores $S_1, \ldots S_n$, where the only requirement is that $S_i = \hat{s}(X_i, Y_i)$ for $i \in [\![n]\!]$, where $\hat{s}$ depends on the observations only via $\mathcal{D}_{\text{train}}$. Typically, the non-conformity score $S_i$ is an increasing function of the distance between $Y_i$ and the prediction at point $X_i$. For example, in a regression framework, the non-conformity scores can be obtained as $S_i = |Y_i - \hat{\mu}(X_i)|$, where $\hat{\mu}(x)$ is a point prediction of $Y_i$ given $X_i = x$ learned on $\mathcal{D}_{\text{train}}$. Alternative non-conformity scores in the same framework include normalized residuals or model-based uncertainties.

Relying on non-conformity scores avoids making assumptions about data distributions or model characteristics. To obtain valid intervals, non-conformity scores have to be exchangeable across the calibration set and test point. Formally, a random vector $(Z_1, \ldots, Z_k)$ is said to be exchangeable (see e.g. Vovk et al., 2005) if for any permutation $\tau \in \mathfrak{S}_k$,

$$(Z_1, \ldots, Z_k) \stackrel{d}{=} (Z_{\tau(1)}, \ldots, Z_{\tau(k)}).$$

Provided that the vector of non-conformity scores $(S_1, \ldots, S_{n+1})$ is exchangeable, SCP yields the following valid interval:

$$\widehat{C}_\alpha = \left[\hat{\mu}\left(X_{n+1}\right) \pm S_{(\lceil (n+1)(1-\alpha)\rceil)}\right].$$

This construction is detailed and thoroughly studied in Papadopoulos et al., 2002; Lei et al., 2018. Alternatively, this interval can also be obtained by thresholding a so-called conformal $p$-value (Vovk et al., 2005; Lei et al., 2018). The conformal $p$-value of test point $(X_{n+1}, Y_{n+1})$ is defined as:

$$p = \frac{1}{n+1}\left(1 + \sum_{j=1}^{n} \mathbf{1}\left\{S_{n+1} \leq S_j\right\}\right) \qquad (2)$$

Intuitively, this $p$-value quantifies how unlikely the non-conformity score $S_{n+1} = \hat{s}(X_{n+1}, Y_{n+1})$ is, given the observed non-conformity scores $(S_i)_{i \in [\![n]\!]}$ computed on the calibration set. In our setting, since $Y_{n+1}$ is unobserved, so are $S_{n+1}$ and the associated conformal $p$-value. This naturally leads to the following definition of a conformal $p$-value function.

**Definition 1** (Conformal $p$-value function, (Barber et al., 2021))**.** *The conformal $p$-value function associated to the score function $\hat{s}$ is $P : \mathbb{R}^p \times \mathbb{R} \to [0, 1]$ with :*

$$P(X, y) = \frac{1}{n+1}\left(1 + \sum_{j=1}^{n} \mathbf{1}\left\{\hat{s}(X_{n+1}, y) \leq \hat{s}(X_j, Y_j)\right\}\right).$$

With this definition, the conformal $p$-value (2) corresponds

to $p = P(X_{n+1}, Y_{n+1})$. In turn, the split conformal prediction confidence interval $\widehat{C}_\alpha$ can be written as

$$\widehat{C}_\alpha(X_{n+1}) = \{y \in \mathbb{R} \text{ s.t. } P(X_{n+1}, y) > \alpha\},$$

which is also sometimes abbreviated as $\widehat{C}_\alpha = \{p > \alpha\}$. With this notation, the coverage property (1) is equivalent to

$$\mathbb{P}\{p \leq \alpha\} \leq \alpha,$$

which is indeed the definition of a valid $p$-value. This duality between confidence intervals and $p$-values in the context of conformal prediction is the basis of the main contributions of the present paper. Reformulating split conformal prediction as a $p$-value thresholding procedure unlocks tools of the rich literature on $p$-value-based error control.

## 2.2. Conformal Prediction for Multiple Test Points

In practical applications, the test set for which we wish to obtain confidence intervals may contain many points. We consider the transductive setting introduced by Vovk (2013), in which $m$ test points $X_{n+1}, \ldots, X_{n+m}$ are observed while the corresponding outcomes $Y_{n+1}, \ldots, Y_{n+m}$ are not observed. Performing split conformal prediction yields $m$ confidence intervals $\mathcal{C}(\alpha) = (\widehat{C}_{i,\alpha})_{i \in [\![m]\!]}$ with

$$\widehat{C}_{i,\alpha} = \left[\hat{\mu}\left(X_{n+i}\right) \pm S_{(\lceil(n+1)(1-\alpha)\rceil)}\right]. \quad (3)$$

Split conformal inference for multiple test points relies on the assumption of exchangeability of the vector of non-conformity scores:

$$(S_i)_{i \in [\![n+m]\!]} \text{is exchangeable} \quad \text{(Exch)}$$

Under (Exch), the following marginal guarantee holds:

$$\forall i \in [\![m]\!], \quad \mathbb{P}\left\{Y_{n+i} \in \widehat{C}_{i,\alpha}\left(X_{n+i}\right)\right\} \geq 1 - \alpha. \quad (4)$$

To provide a quantitative measure of coverage on the set of $m$ points using intervals $\mathcal{I} = (\mathcal{I}_i)_{i \in [\![m]\!]}$, we define the False Coverage Proportion (FCP) and False Coverage Rate (FCR):

$$\text{FCP}(\mathcal{I}) := \frac{1}{m} \sum_{i=1}^{m} \mathbf{1}\left\{Y_{n+i} \notin \mathcal{I}_i\right\},$$

$$\text{FCR}(\mathcal{I}) := \mathbb{E}[\text{FCP}(\mathcal{I})].$$

By (4), FCR control holds at level $\alpha$ for standard split conformal prediction:

$$\text{FCR}(\mathcal{C}(\alpha)) = \frac{1}{m} \sum_{i=1}^{m} \mathbb{P}\left\{Y_{n+i} \notin \widehat{C}_{i,\alpha}(X_{n+i})\right\} \leq \alpha.$$

However, this does not guarantee that $\text{FCP}(\mathcal{C}(\alpha)) \leq \alpha$ with high probability, as noted by Gazin et al., 2024. This is akin to the distinction between False Discovery Proportion control and False Discovery Rate control highlighted in the multiple testing literature (Genovese & Wasserman, 2006).

To ensure informative and interpretable control, our aim is to build FCP upper bounds that hold with high probability. For any pre-specified level $\delta > 0$, this amounts to building $\left(\overline{\text{FCP}}_{\alpha,\delta}\right)_{\alpha \in [0,1]}$ such that:

$$\mathbb{P}\left(\forall \alpha \in [0,1], \quad \text{FCP}(\mathcal{C}(\alpha)) \leq \overline{\text{FCP}}_{\alpha,\delta}\right) \geq 1 - \delta. \quad (5)$$

## 3. Tight FCP Control for Conformal Prediction

### 3.1. FCP Control and Conformal p-values

In this section, we recall the approach introduced by Gazin et al. (2024) to obtain FCP control. This approach stems from a close link between FCP control and the empirical Cumulative Distribution Function (CDF) of conformal $p$-values. Given non-conformity scores $(S_i)_{i \in [\![n+m]\!]}$ the conformal $p$-value of the test point $(X_{n+i}, Y_{n+i})$ for $1 \leq i \leq m$ is defined as:

$$p_i := \frac{1}{n+1}\left(1 + \sum_{j=1}^{n} \mathbf{1}\left\{S_{n+i} \leq S_j\right\}\right).$$

Note that these $p$-values are unobserved, because the definition of $p_i$ involves the unobserved value $Y_{n+i}$ via the corresponding non-conformity score $S_{n+i}$. These $p$-values can also be written as $p_i = P(X_{n+i}, Y_{n+i})$, where $P$ is the conformal $p$-value function introduced in Definition 1. The SCP confidence intervals can then be written as

$$\widehat{C}_{i,\alpha} = \{p_i \leq \alpha\}.$$

**Lemma 1** (Empirical CDF of $p$-values and FCP, Gazin et al., 2024). *Denote by $\widehat{F}_m$ the empirical* CDF *of the joint distribution of $(p_1, \ldots, p_m)$. For any $\alpha \in [0,1]$ denote by $\mathcal{C}(\alpha)$ the split conformal intervals. Then:*

$$\text{FCP}(\mathcal{C}(\alpha)) = \widehat{F}_m(\alpha).$$

Lemma 1 implies that the FCP only depends on the joint distribution of the conformal $p$-values. A key result of Gazin et al., 2024 is the characterization of this distribution under (Exch). This characterization involves the definition of a discrete distribution denoted by $P^U$ on $\left\{\frac{\ell}{n+1}, \ell \in [\![n+1]\!]\right\}$ associated with any fixed vector $U = (U_1, \ldots, U_n) \in [0,1]^n$, as follows:

$$P^U(\{\ell/(n+1)\}) = U_{(\ell)} - U_{(\ell-1)}, \quad \ell \in [\![n+1]\!], \quad (6)$$

where $0 = U_{(0)} \leq U_{(1)} \leq \cdots \leq U_{(n)} \leq U_{(n+1)} = 1$.

**Proposition 1** (Joint distribution of conformal $p$-values, Gazin et al., 2024)**.** *Assume that the non-conformity score vector* $(S_i)_{i \in [\![n+m]\!]}$ *satisfies* (Exch) *and has no ties. Then the* $m$ *conformal* $p$-values *are distributed as* $(q_1, \ldots, q_m)$*, where* $(q_1, \ldots, q_m \mid U) \overset{i.i.d.}{\sim} P^U$ *, with* $U = (U_1, \ldots, U_n) \overset{i.i.d.}{\sim} \mathcal{U}([0,1])$.

The approach of Gazin et al., 2024 to FCP control consists in using this distribution to obtain a Dvoretzky–Kiefer–Wolfowitz–Massart like inequality (DKWM; Massart, 1990). The original DWKM inequality only holds under independence and therefore cannot be used in this context. The obtained inequality bounds the gap between the empirical CDF and the true CDF with high probability. FCP bounds can in turn be obtained using the CDF formulation of Lemma 1.

### 3.2. FCP and Joint Error Rate Control

To obtain FCP control, we instead rely on Joint Error Rate control as introduced by Blanchard et al. (2020). For $k_{max} \in [\![m]\!]$, we define a *threshold family* of size $k_{max}$ as a vector $\mathbf{t} = (t_j)_{j \in [\![k_{max}]\!]}$ such that $0 \leq t_1 \leq \cdots \leq t_{k_{max}} \leq 1$. The Joint Error Rate (JER) is formally defined as follows:

**Definition 2** (Joint Error Rate, Blanchard et al., 2020)**.** *Denote by* $p_{(j)}$ *the* $j^{th}$ *smallest value among* $(p_1, \ldots, p_m)$*. The JER associated with* $\mathbf{t} = (t_j)_{j \in [\![m]\!]}$ *is:*

$$\text{JER}(\mathbf{t}) = \mathbb{P}\left(\exists j \in [\![k_{max}]\!] : p_{(j)} < t_j\right). \quad (7)$$

*The threshold family* $\mathbf{t}$ *is said to control the JER at level* $\delta$ *if* $\text{JER}(\mathbf{t}) \leq \delta$.

A threshold family $\mathbf{t}$ is a (vector) parameter, which has to be chosen in order to provide JER control. In the multiple hypothesis testing literature, JER control has been successfully used to derive bounds on the False Discovery Proportion in various settings (Blanchard et al., 2020; Blain et al., 2022; Durand et al., 2020). Here, we intend to obtain JER control for conformal $p$-values and derive FCP bounds from this control. We first derive from Lemma 1 a tight link between FCP control and JER control.

**Proposition 2** (FCP and JER)**.** *Let* $\mathbf{t}$ *be an arbitrary threshold family. Denote* $j_0(\alpha) = \min\{j \in [\![m]\!] : \alpha \leq t_j\}$ *for* $\alpha \in [0,1]$*. Then:*

$$\mathbb{P}\left(\exists \alpha \in [0,1], \quad \text{FCP}(\mathcal{C}(\alpha)) > \frac{j_0(\alpha)}{m}\right) \leq \text{JER}(\mathbf{t}).$$

*Proof.* We write $\text{JER}(\mathbf{t})$ as a function of $\widehat{F}_m$:

$$\text{JER}(\mathbf{t}) = \mathbb{P}\left(\exists j \in [\![m]\!] : p_{(j)} < t_j\right)$$
$$= \mathbb{P}\left(\exists j \in [\![m]\!] : \sum_{i=1}^{m} \mathbf{1}\{p_i \leq t_j\} \geq j\right)$$
$$= \mathbb{P}\left(\exists j \in [\![m]\!] : \widehat{F}_m(t_j) \geq \frac{j}{m}\right).$$

Then, we note that if for some $\alpha \in [0,1]$ we have $\text{FCP}(\mathcal{C}(\alpha)) \geq j_0(\alpha)/m$, then by definition of $j_0(\alpha)$ combined with Lemma 1, $j := j_0(\alpha) \in [\![m]\!]$ is such that $\widehat{F}_m(t_j) \geq \widehat{F}_m(\alpha) = \text{FCP}(\mathcal{C}(\alpha)) \geq j/m$. $\qquad\square$

As an immediate consequence of Proposition 2, bounds of the form (5) with $\overline{\text{FCP}}_{\alpha,\delta} = j_0(\alpha)/m$ are obtained directly from JER controlling families. Indeed, assuming that $\mathbf{t}$ controls the JER at level $\delta > 0$, then by Proposition 2 the associated $j_0^\delta(\alpha) = \min\{j \in [\![m]\!] : \alpha \leq t_j\}$ satisfies:

$$\mathbb{P}\left(\forall \alpha \in [0,1], \quad \text{FCP}(\mathcal{C}(\alpha)) \leq \frac{j_0^\delta(\alpha)}{m}\right) \geq 1 - \delta. \quad (8)$$

Note that the dependence of $j_0^\delta(\alpha)$ in $\delta$ comes from the fact that $\mathbf{t}$ is assumed to control JER at level $\delta$. As a consequence of the above results, we obtain confidence intervals with uniform in-probability FCP control associated to any on a JER controlling family.

**Corollary 1.** *Assume that* $\mathbf{t}$ *controls the JER at level* $\delta > 0$*. For* $\alpha \in [0,1]$*, let* $\tilde{\alpha} = t_{\lfloor \alpha m \rfloor}$*. Then the associated intervals* $\mathcal{C}(\tilde{\alpha}) = (\widehat{C}_{i,\tilde{\alpha}})_{i \in [[m]]}$ *with* $\widehat{C}_{i,\tilde{\alpha}} = \left[\hat{\mu}\left(X_{n+i}\right) \pm S_{(\lceil (n+1)(1-\tilde{\alpha}) \rceil)}\right]$ *satisfy:*

$$\mathbb{P}\left(\forall \alpha \in (0,1), \text{FCP}(\mathcal{C}(\tilde{\alpha})) \leq \alpha\right) \geq 1 - \delta.$$

To prove Corollary 1, it is sufficient to note that the choice $\tilde{\alpha} = t_{\lfloor \alpha m \rfloor}$ in (8) yields $j_0^\delta(\tilde{\alpha})/m = \lfloor \alpha m \rfloor/m \leq \alpha$. Corollary 1 states that the intervals $\mathcal{C}(\tilde{\alpha})$ have FCP below $\alpha$ with high probability, uniformly in $\alpha$. In order to build these intervals in practice, it only remains to obtain a JER controlling family $\mathbf{t}$.

### 3.3. Building a JER Controlling Family

We follow the approach of Blain et al., 2023, which have exploited related ideas to reach False Discovery control in for Knockoff inference (Candès et al., 2018). As already noted, we can view the threshold family $\mathbf{t}$ as a tuning parameter for JER control. In a nutshell, we first provide an algorithm to estimate the JER associated to a given $\mathbf{t}$, and then use this algorithm to choose a family $\mathbf{t}$ achieving the tightest possible JER control among a set of candidates (which will be called a template).

We start by introducing Algorithm 1, which uses Proposition 1 to draw $B$ Monte-Carlo samples of the joint distribution of conformal $p$-values, and sorts each of them to obtain order statistics. These order statistics are denoted by

---

**Algorithm 1** Sampling order statistics of conformal $p$-**values** using Proposition 1.

1: **Input:** $B$ the number of MC draws; $n$ the number of calibration points; $m$ the number of test points
2: **Output:** $\mathbf{\Pi}_0 \in [0,1]^{B \times m}$ a matrix of simulated $p$-values
3: $\mathbf{\Pi}_0 \leftarrow \text{zeros}(B, m)$
4: **for** $b \in \{1, \dots, B\}$ **do**
5:     Sample $U = [u_1, \dots, u_n]$ from $\mathcal{U}([0,1])^n$
6:     Sample $[q_1, \dots, q_m]$ from $P^U$ **// using** (6)
7:     $\mathbf{\Pi}_0[b] \leftarrow [q_1, \dots, q_m]$
8: **end for**
9: $\mathbf{\Pi}_0 \leftarrow \text{line\_sort}(\mathbf{\Pi}_0)$
10: **Return** $\mathbf{\Pi}_0$

---

$p_{b(1)} \leq \cdots \leq p_{b(m)}$ for each $b \in [\![B]\!]$. This allows us to evaluate the empirical JER, which estimates the actual JER.

**Definition 3** (Empirical JER). *Let $\mathbf{t}$ be a threshold family of size $k_{max}$. The empirical JER of $\mathbf{t}$ associated to a set of $B$ samples of order statistics of conformal $p$-values (as obtained from Algorithm 1) is defined as:*

$$\widehat{\text{JER}}_B(\mathbf{t}) = \frac{1}{B} \sum_{b=1}^{B} \mathbf{1} \left\{ \exists k \in [\![k_{max}]\!] : p_{b(k)} < t_k \right\}.$$

In practice, we compute the empirical JER using Algorithm 2 (Appendix B). In order to ensure JER control, it remains to choose $\mathbf{t}$ such that $\widehat{\text{JER}}(\mathbf{t}) \leq \delta$. To this end, we consider a sorted set of candidate threshold families called a *template*:

**Definition 4** (Template (Blanchard et al., 2020)). *A template is a component-wise non-decreasing function $\mathbf{T}$ : $[0,1] \mapsto \mathbb{R}^p$ that maps a parameter $\lambda \in [0,1]$ to a threshold family $\mathbf{T}(\lambda) \in \mathbb{R}^p$.*

*This definition is naturally extended to the case of templates containing a finite number of threshold families. The template corresponding to $B'$ threshold families is then denoted by $(\mathbf{T}(b'/B'))_{b' \in [\![B']\!]}$.*

Once a template is specified, the *calibration* procedure (Blanchard et al., 2020) can be performed. This consists in finding, among the template, the least conservative threshold family $\mathbf{t}$ that controls the empirical JER at level $\delta$. Formally, we consider the threshold family defined by

$\mathbf{t}_\delta^B = \mathbf{T}(\lambda_B(\delta))$, where

$$\lambda_B(\delta) = \frac{1}{B'} \max \left\{ b' \in [\![B']\!] \text{ s.t. } \widehat{\text{JER}}_B \left( \mathbf{T} \left( \frac{b'}{B'} \right) \right) \leq \delta \right\}.$$

Gazin et al. (2024) also discuss (in Appendix B) possible *a priori* choices for template families in order to obtain FCP control. Here, we propose to *derive the template from the joint distribution of conformal $p$-values itself*. Indeed, as observed by Blain et al. (2022), optimal power is reached when the candidate families match the shape of the distribution of the $p$-values under exchangeability. Therefore, we define a template based on the distribution of the conformal $p$-values appearing in Proposition 1. In practice, we apply Algorithm 1 to obtain a $B' \times m$ matrix of $B'$ samples or order statistics of conformal $p$-values, independently from the first $B$ Monte Carlo samples in order to avoid circularity biases. By sorting each column of this matrix, we obtain a $B' \times m$ matrix $\mathbf{Q}$ of empirical quantiles of these order statistics. This defines a discrete template $\mathbf{T}^0$ in the sense of Definition 4, where $\mathbf{T}^0 (b'/B')$ is the $b'$-th row of $\mathbf{Q}$ for $b' \in [\![B']\!]$. This construction is summarized in Algorithm 3 (Appendix B). We obtain the following result, which is akin to Theorem 2 in Blain et al. (2023):

**Proposition 3** (JER control for conformal $p$-values). *Consider the threshold family defined by $\mathbf{t}_\delta^B = \mathbf{T}^0(\lambda_B(\delta))$. Then, as $B \to +\infty$,*

$$\text{JER}(\mathbf{t}_\delta^B) \leq \delta + O_P(1/\sqrt{B}).$$

Proposition 3 is a consequence of the fact that $\widehat{\text{JER}}_B(\mathbf{t}_\delta^B) \leq \delta$ (which holds by the definition of $\mathbf{t}_\delta^B$), combined with the fact that the difference between $\text{JER}_B$ and its Monte-Carlo approximation $\widehat{\text{JER}}_B$ is uniformly bounded in probability by $1/\sqrt{B}$ as $B \to +\infty$. The proof of Proposition 3 is given in Appendix A for completeness.

The number $B$ of Monte-Carlo samples in Proposition 3 can be chosen arbitrarily large to obtain JER control. This leads to a valid FCP bound by Corollary 1. Algorithm 3 describes all the steps needed to compute $\mathbf{t}_\delta^B$. The resulting FCP bound is fully nonparametric and therefore expected to yield tighter intervals than Gazin et al., 2024's approach. We call the resulting approach **CoJER** (Conformal - JER).

## 4. Aggregated Conformal Prediction

While conformal prediction coverage guarantees are *distribution free*, the confidence interval output by the method can strongly depend on the chosen model $\hat{\mu}$ in practice. Mitigating the consequences of such modeling decisions motivates the use of aggregation schemes to obtain more stable and generalizable confidence intervals.

Let us assume that we have $K$ models $\hat{\mu}_1, \dots, \hat{\mu}_K$ fitted on $\mathcal{D}_{train}$. The goal of aggregation is to build a valid confi-

dence interval $\widehat{C}(\alpha)$ that takes into account the information provided by each model. Aggregating schemes for conformal prediction have been introduced in Lei et al., 2018; Barber et al., 2021. Lei et al., 2018 propose a Bonferroni-type construction, where the confidence interval of each of the $K$ models is built at level $\alpha/K$. An union bound argument shows that the intersection of these intervals is valid at level $\alpha$, therefore yielding FCR control at level $\alpha$.

Barber et al., 2021 propose a method that relies on a $p$-value aggregation result which states that twice the arithmetic mean of valid $p$-values is a valid $p$-value – see e.g. Vovk & Wang (2020). This results in a FCR controlling procedure. Therefore, existing solutions require the construction of *valid* aggregated $p$-values. Here, we define a generic aggregation scheme that yields FCP control using the same tools as in Blain et al., 2023. In particular, our proposed construction does not require the result of aggregation to be a valid $p$-value. First, let us define an aggregation procedure:

**Definition 5.** *Given $K$ models, an aggregation procedure is a function $f : \mathbb{R}^K \mapsto \mathbb{R}$ that maps a vector of $(p^k)_{k \in [\![K]\!]}$ statistics to a scalar statistic $\overline{p}$.*

In practice, since we have $m$ test points, aggregation is performed for each test point, i.e.:

$$\forall j \in [\![m]\!], \quad \overline{p_j} = f(p_j^1, \ldots, p_j^K).$$

Then, inference is performed on the vector of aggregated $p$-values $(\overline{p}_1, \ldots, \overline{p}_m)$.

For a fixed aggregation scheme $f$, we can naturally extend the calibration procedure of the preceding section. Instead of drawing a single $B \times p$ matrix of conformal $p$-values containing $p_b \in \mathbb{R}^p$ for each $b \in [\![B]\!]$, we draw $K$ such matrices. Given $k \in [\![K]\!]$, each matrix contains $p_b^k \in \mathbb{R}^p$ for each $b \in [\![B]\!]$. We then perform aggregation: $\overline{p}_b = f\left((p_b^k)_{k \in [\![K]\!]}\right)$. The JER in the aggregated case is defined as:

$$\overline{\mathrm{JER}}(\mathbf{t}) = \mathbb{P}\left(\exists j \in [\![k_{max}]\!] : \overline{p}_{(j)} < t_j\right).$$

We obtain the aggregated template following the same procedure, i.e. drawing $K$ templates and aggregating them. For each $b' \in [\![B']\!]$, the aggregated threshold family is written:

$$\overline{\mathbf{T}}\left(\frac{b'}{B'}\right) = f\left(\left(\mathbf{T}^k\left(\frac{b'}{B'}\right)\right)_{k \in [\![K]\!]}\right).$$

We can then write the empirical JER in the aggregated case as:

$$\widehat{\overline{\mathrm{JER}}}\left(\overline{\mathbf{T}}\left(\frac{b'}{B'}\right)\right) = \frac{1}{B}\sum_{b=1}^{B} \mathbf{1}\left\{\exists j \in [\![k_{max}]\!] : \overline{p}_{b(j)} < \overline{\mathbf{T}}_j\left(\frac{b'}{B'}\right)\right\}.$$

Calibration can be performed in the same way as in the non-aggregated case. Note that we perform calibration *after* aggregation; therefore, JER control is ensured directly on aggregated $p$-values and is not a result of aggregating JER controlling families. Importantly, this approach holds without additional assumptions on the aggregation scheme $f$. In particular, contrary to the approaches of Lei et al., 2018; Barber et al., 2021, our aggregated $p$-values are not required to be valid $p$-values for the corresponding intersection hypothesis. We consider the threshold family $\overline{\mathbf{t}}_\delta^B = \overline{\mathbf{T}}(\lambda_B(\delta))$, where

$$\lambda_B(\delta) = \frac{1}{B'} \max\left\{b' \in [\![B']\!] \text{ s.t. } \widehat{\overline{\mathrm{JER}}}_B\left(\overline{\mathbf{T}}\left(\frac{b'}{B'}\right)\right) \leq \delta\right\}.$$

With $\overline{\mathbf{T}}^0$ a template composed of $B'$ candidate curves that match quantiles of the distribution of aggregated conformal $p$-values $\overline{p}$, we obtain the following result:

**Proposition 4** (JER control for aggregated conformal $p$-values)**.** *Consider the threshold family defined by $\overline{\mathbf{t}}_\delta^B = \overline{\mathbf{T}}^0(\lambda_B(\delta))$. Then, as $B \to +\infty$,*

$$\overline{\mathrm{JER}}(\overline{\mathbf{t}}_\delta^B) \leq \delta + O_P(1/\sqrt{B}).$$

*Proof.* The proof is identical to that of Proposition 3 using the empirical aggregated JER. $\square$

The calibrated aggregated threshold family yields valid FCP upper bounds via Corollary 1. We therefore achieve a fully nonparametric aggregation scheme for conformal prediction, along with guarantees on the FCP.

## 5. Experiments

**Setup.** We use 17 OpenML (Vanschoren et al., 2014) datasets from (Grinsztajn et al., 2022). Each dataset is randomly split ($n_{split} = 30$ times) into a train, calibration and test set. The latter is of size $m$ and denoted by $\mathcal{D}_{test}^s$. We fit 5 regression models on the training sets[1]: Random Forest (RF) (Breiman, 2001), Multi-Layer Perceptron (MLP) (Hinton, 1990), Support Vector Regression (SVR) (Platt et al., 1999) K-Nearest Neighbors (KNN; Cover & Hart, 1967) and Lasso (Tibshirani, 1996).

**FCP control.** We consider three methods for comparison: classical Split Conformal Prediction, the method proposed by Gazin et al., 2024 to obtain FCP control via DKW-type bounds (Massart, 1990) and the proposed approach. We use $\alpha = 0.1$ for all methods. For FCP controlling methods, we set $\delta = 0.1$ and use SCP with the largest level $\alpha'$ such that $\overline{\mathrm{FCP}}_{\alpha', \delta} \leq \alpha$.

---

[1]All experiments were performed using 40 CPUs, Intel(R) Xeon(R) CPU E5-2660 v2 @ 2.20GHz

For each dataset, we compute for each split the empirical FCP for each model and conformal prediction method. Formally, for a given data set, denoting by $\mathcal{C}^s = \left( \widehat{C}_i^s \right)_{i \in \mathcal{D}_{test}^s}$ the confidence intervals obtained for the $s$-th split for a given method, the associated empirical FCP is given by:

$$\mathrm{FCP}(\mathcal{C}^s) = \frac{1}{m} \sum_{i \in \mathcal{D}_{test}^s} \mathbf{1} \left\{ Y_i \notin \widehat{C}_i^s \right\}.$$

Then for each dataset, we compute the associated empirical coverage as the proportion of splits for which the FCP control event holds:

$$\frac{1}{n_{splits}} \sum_{s=1}^{n_{splits}} \mathbf{1} \left\{ \mathrm{FCP}(\mathcal{C}^s) < \alpha \right\}.$$

We also compute the interval length of each method for each dataset. We report the relative length to the shortest interval found among all methods, averaged across all splits for each dataset. This allows having a comparable metric for interval informativeness across all datasets.

The left panel of Figure 1 shows that across all models and datasets, standard Split Conformal does not guarantee FCP control at level $\alpha$ – this is consistent with theory, as Split Conformal prediction only guarantees FCR control. Strikingly, the proportion of splits for which $\mathrm{FCP} \leq \alpha$ for Split Conformal can be as low as $35\%$ for certain models and datasets. Both the proposed method and the method of Gazin et al., 2024 control the FCP as expected. Concretely, this means that for all datasets, the proportion of splits for which $\mathrm{FCP} \leq \alpha$ is indeed superior to $1 - \delta$.

The right panel of Figure 1 shows that SCP yields the shortest intervals in all settings. This is expected, as FCR control is less stringent than FCP control, leading to shorter intervals. Among the two FCP controlling methods, the proposed method is less conservative than the method of Gazin et al., 2024. On average across all models and datasets, the proposed method yields intervals that are only $\sim 15\%$ larger than standard SCP. In worst-case scenarios, the proposed method yields intervals $\sim 25\%$ larger than SCP, while intervals obtained using the method of Gazin et al., 2024 are $\sim 80\%$ larger than SCP intervals. Overall, the proposed method yields sharp FCP control at a modest cost in terms of interval length compared to SCP.

**Aggregation.** We use the five regression models mentioned above and consider three aggregation methods for comparison: the method based on the arithmetic mean of $p$-value functions proposed by Barber et al., 2021 labeled

CV+, the Bonferroni-like construction of Lei et al., 2018 and the proposed method. For the proposed method, we use the harmonic mean as the aggregation scheme. In Appendix C we provide a comparison to other possible aggregation schemes (arithmetic mean, geometric mean and quantile aggregation). This comparison shows that the harmonic mean consistently outperformed the other aggregation schemes considered in terms of interval tightness.

As in the first experiment, we compute the FCP coverage of each method and the relative interval length. The left panel of Figure 2 shows that all three methods control the FCP at level $\delta = 0.1$. While CoJER offers a theoretical guarantee on this control, this is not the case for CV+ and Bonferroni. These two methods likely control the FCP due to excessive conservativeness, as the FCP event is controlled $100\%$ of the time for most datasets using CV+ and Bonferroni intersection.

The right panel of Figure 2 shows that CoJER yields the most informative intervals across all datasets. The intervals yielded by the CV+ procedure are $\sim 75\%$ larger on average than those of CoJER. The Bonferroni-intersection intervals are $\sim 20\%$ larger on average than those of CoJER. Overall, these experiments show that the proposed nonparametric aggregation scheme achieves sharp FCP control while providing tighter intervals that *state-of-the-art* methods.

## 6. Discussion

In this paper, we have proposed a novel method that allows sharp control of the false coverage proportion in conformal prediction. This method builds upon JER control introduced by (Blanchard et al., 2020) and on a characterization of the joint distribution of conformal $p$-values recently established by (Gazin et al., 2024). Our main contribution of this paper with respect to these papers is twofold. First, we build tighter FCP confidence bounds than (Gazin et al., 2024) by deriving the shape of the template family from the distribution of conformal $p$-values itself. Second, we introduce an aggregation scheme which yields an integrated method which provides robustness with respect to modeling choices, and yields tighter intervals than other existing aggregation schemes.

The computational cost of this method is comparable to classical SCP. For given sizes of calibration and test sets, sampling conformal $p$-values from Algorithm 1 can be done once and for all. Calibration using Algorithm 3 is performed via binary search of complexity $\mathcal{O}(log(B'))$. Computing the empirical JER of a threshold family using Algorithm 2 has a computational complexity of $\mathcal{O}(Bk_{max})$.

Additionally, once calibration is performed, the bound of Corollary 1 holds simultaneously for all values of $\alpha$. In practice, users can try different values of $\alpha$ *post hoc* while

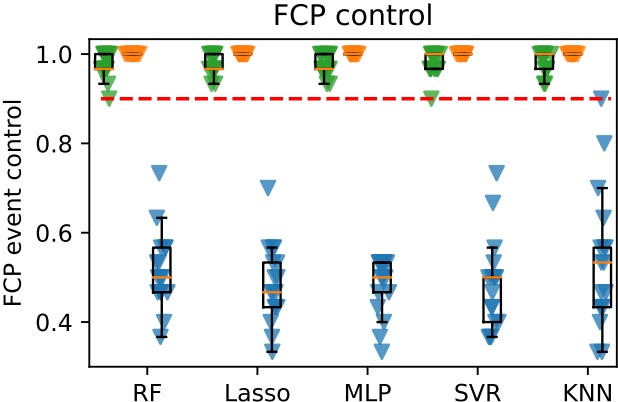
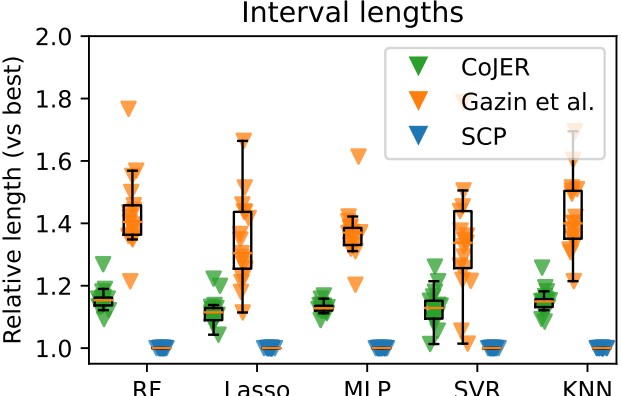

Figure 1: **Coverage and relative interval length for 5 models.** We use 17 OpenML (Vanschoren et al., 2014) datasets from (Grinsztajn et al., 2022). Each dataset is split (30 times) into a train, calibration and test set. We fit 5 regression models on the training sets: Random Forest (RF), Multi-Layer Perceptron (MLP), Support Vector Regression (SVR), K-Nearest Neighbors (KNN) and Lasso. Calibration sets are used to compute SCP intervals and conformal $p$-values. For each method and dataset, we report the FCP coverage, i.e. the proportion of test splits for which the event FCP $\leq \alpha$ was realized. We also report the interval length, relative to the smallest valid interval found among all methods. Notice that standard SCP does not guarantee FCP control at level $\alpha$: for certain datasets and models, FCP event coverage can be as low as 30%. Both the proposed approach and Gazin et al., 2024 obtain FCP control. However, the proposed approach is much less conservative.

retaining valid FCP bounds without needing to relaunch the complete procedure.

Notably, valid $p$-values are not needed to obtain FCP control. Since this control is a direct consequence of JER control on aggregated $p$-values, it can be obtained for any aggregation scheme $f$. In particular, our use of the harmonic mean to aggregate $p$-values leads to valid FCP control, even if the harmonic mean does not always yield valid $p$-values (Chen et al., 2024).

While we have focused on the regression setting for the numerical experiments reported in this paper, our proposed method inherits the genericity of SCP and could thus also be applied to classification tasks. This method could also be extended to other uncertainty quantification frameworks such as bootstrap or resampling based methods like the jackknife+ (Barber et al., 2021). Since valid $p$-values are not needed, characterizing the distribution of statistics quantifying uncertainty is sufficient to apply the proposed method.

Another interesting avenue of work is to study the impact of controlling the FCP rather than the FCR for downstream decisions taken using confidence intervals. This type of analysis has been conducted in Vovk & Bendtsen, 2018 in the classical conformal prediction framework and in Perez-Lebel et al., 2024 in the field of model calibration.

An implementation of CoJER is available at https:// github.com/sanssouci-org/CoJER-paper, together with the code to reproduce the numerical results of this paper.

## Acknowledgments

This research has received funding from the KARAIB AI chair (ANR-20-CHIA-0025-01), the H2020 Research Infrastructures Grant EBRAIN-Health 101058516, and the VITE ANR-23-CE23-0016 Grant.

## Impact Statement

This paper presents work whose goal is to advance the field of Machine Learning. There are many potential societal consequences of our work, none which we feel must be specifically highlighted here.

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

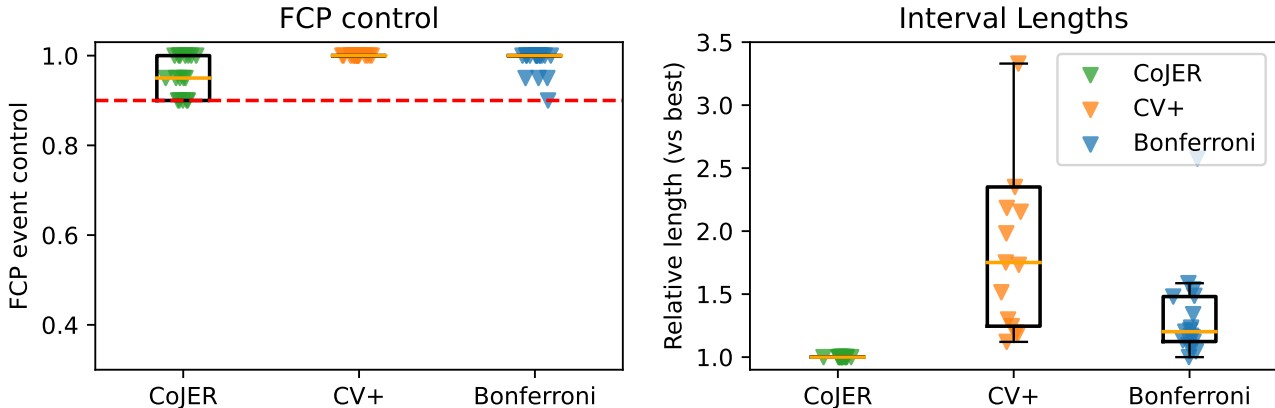

Figure 2: **Coverage and relative interval length for aggregation-based methods using 5 models.** We use 17 OpenML (Vanschoren et al., 2014) datasets from (Grinsztajn et al., 2022). Each dataset is split into a train, calibration and test set. We fit 5 regression models on the training sets: Random Forests, Multi-Layer Perceptron, Support Vector Regression, K-Nearest Neighbors and Lasso. Calibration sets are used to compute Split Conformal prediction intervals and conformal $p$-value functions. These functions are used to compute confidence intervals for the proposed method. For each method and dataset, we report the FCP coverage, i.e. the proportion of test splits for which the event FCP $\leq \alpha$ was realized. We also report the interval length, relative to the smallest valid interval found among all methods. The proposed method achieves the expected FCP coverage while providing the most informative intervals.

discovery proportion control for aggregated knockoffs. *NeurIPS 2023*, 2023.

Blanchard, G., Neuvial, P., Roquain, E., et al. Post hoc confidence bounds on false positives using reference families. *Annals of Statistics*, 48(3):1281–1303, 2020.

Breiman, L. Random forests. *Machine learning*, 45:5–32, 2001.

Candès, E., Fan, Y., Janson, L., and Lv, J. Panning for gold:'model-x'knockoffs for high dimensional controlled variable selection. *Journal of the Royal Statistical Society: Series B (Statistical Methodology)*, 80(3): 551–577, 2018.

Chen, Y., Wang, R., Wang, Y., and Zhu, W. Sub-uniformity of harmonic mean p-values. *arXiv preprint arXiv:2405.01368*, 2024.

Cover, T. and Hart, P. Nearest neighbor pattern classification. *IEEE transactions on information theory*, 13(1): 21–27, 1967.

Durand, G., Blanchard, G., Neuvial, P., and Roquain, E. Post hoc false positive control for structured hypotheses. *Scandinavian journal of Statistics*, 47(4):1114–1148, 2020.

Gazin, U., Blanchard, G., and Roquain, E. Transductive conformal inference with adaptive scores. In *International Conference on Artificial Intelligence and Statistics*, pp. 1504–1512. PMLR, 2024.

Genovese, C. R. and Wasserman, L. Exceedance control of the false discovery proportion. *Journal of the American Statistical Association*, 101(476):1408–1417, 2006.

Grinsztajn, L., Oyallon, E., and Varoquaux, G. Why do tree-based models still outperform deep learning on typical tabular data? *Advances in neural information processing systems*, 35:507–520, 2022.

Hinton, G. E. Connectionist learning procedures. In *Machine learning*, pp. 555–610. Elsevier, 1990.

Lei, J., G'Sell, M., Rinaldo, A., Tibshirani, R. J., and Wasserman, L. Distribution-free predictive inference for regression. *Journal of the American Statistical Association*, 113(523):1094–1111, 2018.

Massart, P. The tight constant in the dvoretzky-kiefer-wolfowitz inequality. *The annals of Probability*, pp. 1269–1283, 1990.

Meinshausen, N., Meier, L., and Bühlmann, P. P-values for high-dimensional regression. *Journal of the American Statistical Association*, 104(488):1671–1681, 2009.

Papadopoulos, H., Proedrou, K., Vovk, V., and Gammerman, A. Inductive confidence machines for regression. In *Machine learning: ECML 2002: 13th European conference on machine learning Helsinki, Finland, August*

*19–23, 2002 proceedings 13*, pp. 345–356. Springer, 2002.

Perez-Lebel, A., Varoquaux, G., Koyejo, S., Doutreligne, M., and Morvan, M. L. Decision from suboptimal classifiers: Regret pre- and post-calibration. *Under Review*, 2024.

Platt, J. et al. Probabilistic outputs for support vector machines and comparisons to regularized likelihood methods. *Advances in large margin classifiers*, 10(3):61–74, 1999.

Tibshirani, R. Regression shrinkage and selection via the lasso. *Journal of the Royal Statistical Society: Series B (Methodological)*, 58(1):267–288, 1996.

Vanschoren, J., Van Rijn, J. N., Bischl, B., and Torgo, L. Openml: networked science in machine learning. *ACM SIGKDD Explorations Newsletter*, 15(2):49–60, 2014.

Vovk, V. Transductive conformal predictors. In *Artificial Intelligence Applications and Innovations: 9th IFIP WG 12.5 International Conference, AIAI 2013, Paphos, Cyprus, September 30–October 2, 2013, Proceedings 9*, pp. 348–360. Springer, 2013.

Vovk, V. and Bendtsen, C. Conformal predictive decision making. In *Conformal and Probabilistic Prediction and Applications*, pp. 52–62. PMLR, 2018.

Vovk, V. and Wang, R. Combining p-values via averaging. *Biometrika*, 107(4):791–808, 2020.

Vovk, V., Gammerman, A., and Shafer, G. *Algorithmic learning in a random world*, volume 29. Springer, 2005.

## A. Proofs

**Lemma 2.** *For any threshold family* $\mathbf{t}$*, we have*

$$\mathrm{JER}\left(\mathbf{t}\right) - \widehat{\mathrm{JER}}_B(\mathbf{t}) = O_P(1/\sqrt{B})$$

*Proof of Lemma 2.* Let $Z_B(\mathbf{t}) = \sqrt{B}\left(\mathrm{JER}\left(\mathbf{t}\right) - \widehat{\mathrm{JER}}_B(\mathbf{t})\right)$. By the Central Limit Theorem, we have

$$Z_B(\mathbf{t}) \xrightarrow[B\to\infty]{d} Z(\mathbf{t}),$$

where $Z(\mathbf{t})$ is a centered Gaussian random variable with variance $\sigma^2(\mathbf{t}) = \mathrm{JER}\left(\mathbf{t}\right)(1 - \mathrm{JER}\left(\mathbf{t}\right))$. As such, for any $M > 0$, we have

$$\mathbb{P}\left(|Z_B(\mathbf{t})| \geq M\right) \xrightarrow[B\to\infty]{} \mathbb{P}\left(|Z(\mathbf{t})| \geq M\right).$$

Since $\mathrm{JER}^0\left(\mathbf{t}\right) \leq 1$, we have $\sigma^2(\mathbf{t}) \leq 1/4$ for any $\mathbf{t}$, so that $Z(\mathbf{t})$ is stochastically dominated by $\mathcal{N}\left(0, 1/4\right)$, which does not depend on the threshold family $\mathbf{t}$. As such, we have $\mathbb{P}\left(|Z(\mathbf{t})| \geq M\right) = 2\mathbb{P}\left(Z(\mathbf{t}) \geq M\right) \leq 2\overline{\Phi}(2M)$, where $\overline{\Phi}$ denotes the tail function of the standard normal distribution. Since $\overline{\Phi}(x)$ tends to 0 as $x \to +\infty$, we have proved that $Z_B(\mathbf{t}) = O_P(1)$. □

**Proposition 3** (JER control for conformal $p$-values)**.** *Consider the threshold family defined by* $\mathbf{t}_\delta^B = \mathbf{T}^0(\lambda_B(\delta))$*. Then, as* $B \to +\infty$*,*

$$\mathrm{JER}(\mathbf{t}_\delta^B) \leq \delta + O_P(1/\sqrt{B}).$$

*Proof.* We treat the case where $\mathbf{t}_\delta^B$ is well defined for all $B$, i.e. that there exists a threshold family among $\mathbf{T}^0$ controls the empirical $\mathrm{JER}^0$ for $B$ draws. If this is not the case for some $B$, then $\mathbf{t}_\delta^B$ is set to the null family and the result holds.

We can write:

$$\mathrm{JER}\left(\mathbf{t}\right) = \widehat{\mathrm{JER}}_B(\mathbf{t}) + \left(\mathrm{JER}\left(\mathbf{t}\right) - \widehat{\mathrm{JER}}_B(\mathbf{t})\right)$$
$$= \widehat{\mathrm{JER}}_B(\mathbf{t}) + O_P(1/\sqrt{B})$$

by Lemma 2. Applying the above to $\mathbf{t} = \mathbf{t}_\delta^B$ yields the desired result since $\widehat{\mathrm{JER}}_B(\mathbf{t}_\delta^B) \leq \delta$ by definition. □

## B. Algorithms

In this section, we provide Algorithm 2 to compute the empirical JER of Definition 3 and Algorithm 3 to perform the calibration procedure on conformal $p$-values.

**Algorithm 2 Computing the Empirical JER.** The empirical JER is computed for a given threshold family and a matrix of conformal $p$-values. This algorithm is similar to Algorithm 3 of (Blain et al., 2022).

1: **Input:** $\Pi$ a matrix of conformal $p$-values; $\mathbf{t}$ a threshold family; $k_{max}$ the size of the threshold family
2: **Output:** $\widehat{\text{JER}}$, the empirical JER of threshold family $\mathbf{t}$
3: $(B, \text{m}) \leftarrow \text{shape}(\Pi)$
4: $\widehat{\text{JER}} \leftarrow 0$
5: **for** $b \in \{1, \ldots, B\}$ **do**
6:    **for** $i \in \{1, \ldots, k_{max}\}$ **do**
7:       $\text{diff}[i] \leftarrow \Pi[b][i] - \mathbf{t}[i]$ **`// Check JER control at rank`** $i$
8:    **end for**
9:    **if** $\min(\text{diff}) < 0$ **then**
10:       $\widehat{\text{JER}} \leftarrow \widehat{\text{JER}} + \frac{1}{B}$ **`// Increment risk if JER control event is violated`**
11:    **end if**
12: **end for**
13: **Return** $\widehat{\text{JER}}$

---

**Algorithm 3 Performing calibration on conformal $p$-values.** First, we use Algorithm 1 to build a suitable template and estimate the JER of each candidate threshold family. Then, we perform calibration to select the least conservative possible threshold family that controls the JER at a given level $\delta$.

1: **Input:** $\delta$ the desired coverage; $B$ the number of MC draws for JER estimation; $B'$ the number of candidate threshold families
2: **Output:** $\mathbf{t}_\delta$ the calibrated threshold family at level $\delta$
3: $\Pi \leftarrow \text{draw\_conformal\_p}(B, m)$ **`// Algo. 1`**
4: $\Pi' \leftarrow \text{draw\_conformal\_p}(B', m)$ **`// Algo. 1`**
5: **for** $b' \in \{1, \ldots, B'\}$ **do**
6:    $\mathbf{Q}[b'] \leftarrow \text{quantiles}(\Pi', b'/B')$ **`// Build template`**
7:    $\widehat{\text{JER}}[b'] \leftarrow \text{empirical\_jer}(\Pi, \mathbf{Q}[b'])$ **`// Apply Algorithm 2 to each family`**
8: **end for**
9: $b'_{cal} \leftarrow \max\{b' \in \{1, \ldots, B'\} \text{ s.t. } \widehat{\text{JER}}[b'] \leq \delta\}$ **`// Perform calibration`**
10: $\mathbf{t}_\delta \leftarrow \mathbf{Q}[b'_{cal}]$
11: **Return** $\mathbf{t}_\delta$

---

## C. Comparison Between Aggregation Schemes

We performed an additional experiment on the 17 OpenML datasets used in Section 5. In this experiment, we compare four possible aggregation schemes: harmonic mean, arithmetic mean, geometric mean and quantile aggregation (Meinshausen et al., 2009).

We use the setup described in Section 5 - i.e. $\alpha = 0.1, \delta = 0.1$. Importantly, we first check that the FCP is controlled for all types of aggregation by reporting the FCP event coverage. This value is expected to be above $1 - \delta = 90\%$. We also compute the (relative) interval width for each aggregation scheme, averaged across 20 splits.

| | FCP event coverage | Interval width increase (vs best) |
|---|---|---|
| Harmonic mean | 94% | **0%** |
| Geometric mean | 100% | +24% |
| Arithmetic mean | 100% | +230% |
| Quantile aggregation | 100% | +54% |

Table 1: Comparing four possible aggregation schemes on 17 OpenML datasets. We report the empirical FCP coverage (for a target coverage of $1 - \delta = 90\%$) and relative interval width for harmonic mean, arithmetic mean, geometric mean and quantile aggregation (Meinshausen et al., 2009). The FCP is controlled for all four aggregation schemes. Harmonic mean aggregation consistently outperforms the other aggregation schemes.

The results are presented in Table 1. Coherently with the theoretical guarantees obtained in Proposition 4, the FCP is controlled for all four aggregation schemes. In terms of interval tightness, harmonic mean aggregation outperforms arithmetic mean, geometric mean and quantile aggregation consistently.

