# OpenReview forum: "False Coverage Proportion Control for Conformal Prediction"
_ICML.cc/2025/Conference — ICML 2025 poster_

### Official Review · Reviewer_ExJ2 · 2025-03-13

**Overall Recommendation:** 3

**Summary:**

The authors propose using the Joint Error Control (JER) framework of Blanchard et al. (2020) to control the false coverage proportion (FCP) across multiple conformal prediction intervals. This approach leverages the exact joint distribution of conformal $p$-values derived in Gavin et al. (2024). They introduce a specific instantiation of the JER method, selecting a particular "template" and "threshold function," which they argue is optimal or tight, though this claim is not formally proven. Additionally, they propose a method for aggregating conformal $p$-values and intervals while ensuring that both the FCP and JER remain controlled. In experiments, the proposed methods outperform uncorrected approaches and existing, more conservative, methods.

**Claims And Evidence:**

The authors claim that their approach is expected to be tighter than that of Gazin et al. (2024). Why is this the case? Can you provide a formal result?

Additionally, Section 1 states that “while the approach of Gazin et al. (2024) yields valid FCP bounds, they are fully parametric, which can entail conservativeness, as discussed by the authors.” However, upon reviewing their paper, they also consider distribution-free conformal prediction settings, and I do not see any explicit dependence on parametric assumptions. Can the authors clarify what they mean by "parametric" in this context and explain why they consider the approach of Gazin et al. (2024) to be conservative?

**Essential References Not Discussed:**

NA

**Experimental Designs Or Analyses:**

The experiments appear sound and comprehensive.

**Methods And Evaluation Criteria:**

NA

**Other Comments Or Suggestions:**

- In Lemma 1, how are \( p_1, \dots, p_m \) defined? Is \( p_j \) equal to \( P(Y_j) \), where \( P(\cdot) \) represents the p-value function?

- The concepts of threshold families and templates may be abstract for those unfamiliar with this literature. Providing some context on why these notions are introduced and their practical applications could be helpful.

-  To facilitate broader adoption and improve accessibility within the CP community, it might be useful to present a less abstract version and intuition of the algorithm, e.g., introduce the method for a simple template and threshold function. Is the general idea to decrease the rejection thresholds of the p-values monotonically, thereby effectively lowering the significance (alpha) levels of the CP intervals? How does this procedure compare to and differ from the Benjamini-Hochberg method, which may be more familiar to readers?

- For the template constructed using Monte Carlo simulation described above in Proposition 3, is this template optimal in power when the number of MC replicates ($B$) approaches infinity? It would be beneficial to formally define the optimal template function, as the informal statement that it must "match the shape of the distribution of the $p$-values under exchangeability" is unclear. Does the proposed template approximate such an oracle template? Formal results on the optimality of the procedure or an oracle variant of it would be appreciated.


- Does the model aggregation scheme preserve the coverage properties of CP? Does this follow from JER control?

- The title does not seem appropriate, as there are no tightness guarantees and "reliability" is vague and CP is arguably reliable as is. The title should reflect the actual contributions of the paper, e.g.,  something like "Controlling the FCP/JER of CP".

**Other Strengths And Weaknesses:**

### Strengths

- The application of the Joint Error Control (JER) framework of Blanchard et al. (2020) to conformal prediction is novel
- The proposed aggregation scheme for conformal $p$-values and intervals that ensures control of bot FCP and JER is novel and practically useful.


### Weaknesses

- It is not explained how calibrated conformal prediction intervals are derived from the calibrated $p$-values and the threshold family. How should the threshold family output by Algorithm 3 be used to construct conformal intervals with JER control? Without explicitly defining these intervals, the utility of Proposition 3 is unclear.  Am I supposed to apply Corollary 1 with the adjusted interval? If so, this interval does not appear to be explicitly defined.


- The writing is somewhat unclear, and the paper could benefit from additional intuition and background on multiple testing methods. For instance, the motivation for introducing threshold families and templates, as well as their utility, may not be immediately clear to readers.


- The theoretical results primarily build on or modify existing work, and some key formal results are missing. Specifically:
  1. There are no optimality properties established for the template proposed for JER control in Section 3.3.
  2. There are no formal results justifying the claims of tightness and sharpness of the proposed intervals compared to existing methods (e.g., Gazin et al., 2024).
  3. There are no formal or informal results demonstrating that the model aggregation method preserves the coverage properties, although JER control is established.  If marginal coverage follows from JER control this should be stated.

**Questions For Authors:**

- Once we have obtained the calibrated p-values using Algorithm 3, how do we derive the adjusted conformal prediction intervals? Are the sequence of \( t \)'s the new alpha values for the corresponding sequence of prediction intervals? Does Algorithm 3 provide a calibrated p-value function that can be used to construct the new interval following the approach in Definition 2? It would be helpful to include an explicit end-to-end algorithm for constructing the calibrated intervals.


- In proposition 2, does the definition of the interval C(alpha) depend on the threshold family? In Section 3.3, it sounds like the idea of the proposed method is to find a threshold family whose JER is controlled at a desired level. However, as mentioned above, it is not clear how this threshold family is mapped to new intervals.

-  It would be beneficial to present a formal result in Section 3.3 on the JER control of the proposed method by combining Proposition 3 with Corollary 1. Specifically, for the method proposed by the authors, what is the value or order of \( j(\alpha, \delta) \)? How does the bound behave when \( \delta = 1/\sqrt{n} \) or \( \delta = 1/n \)?

# Post review
I have raised my score from a 2 to a 3 and am positive of the work.

**Relation To Broader Scientific Literature:**

NA

**Theoretical Claims:**

NA

---

> ### Author Rebuttal · Authors · 2025-03-31
>
> We thank the reviewer for their time and insightful comments. Please find our answers to the points raised below.
>
> >In Lemma 1, how are ( p_1, \dots, p_m ) defined? Is ( p_j ) equal to ( P(Y_j) ), where ( P(\cdot) ) represents the p-value function?
>
> For each $j \in [[m]]$, we define
> $p_j := \frac{1}{n+1} \left(1 + \sum_{i=1}^n \mathbf{1}[S_j \le S_i]\right)$ as per Definition 1. We have added this to Lemma 1 in the updated version of the manuscript.
>
> > The concepts of threshold families and templates may be abstract for those unfamiliar with this literature. Providing some context on why these notions are introduced and their practical applications could be helpful.
>
> We will add a pedagogical figure that explains the procedure in two steps in the updated manuscript -- we detailed this in our answer to reviewer DTau. In a nutshell, the threshold family can be viewed as a (vector) parameter, and templates are sets of candidate parameters, from which a candidate ensuring JER control (and hence FCP coverage) is chosen by the calibration algorithm.
>
> **Regarding the "parametric" nature of the Gazin et al. bound**, we agree that the use of "parametric" can be confusing here. Indeed, the paper of Gazin et al. (2024) considers distribution-free conformal prediction settings. We wanted to emphasize that the **confidence envelope of $\widehat{F}_m$ they propose has a fixed shape**. Theorem 2.3 of Gazin et al. (2024) is a Dvoretzky–Kiefer–Wolfowitz (DKW) type inequality, proved to hold for the specific (known) dependence between conformal $p$-values. As noted in that paper (see Remark 2.6 therein), this approach can be conservative, which can be addressed by the JER calibration framework. CoJER is fully non-parametric in the sense that the shape of the tempate is itself derived from the joint distribution of $p$-values, and not chosen *a priori*. Our experimental results show that CoJER is much less conservative than the approach of Gazin et al. (2024) while preserving FCP control.
>
> **Explicit construction of the confidence intervals:** we thank the reviewer for pointing this out. Once Algorithm 3 is run, we obtain a JER controlling family $t$. Corollary 1 states that, with probability at least $1-\delta$, $\forall \alpha \in[0,1], \quad \operatorname{FCP}(\mathcal{C}(\alpha)) \leq \frac{j(\alpha, \delta)}{m}$ with $j(\alpha, \delta) = \min \{j \in [[m]] : \alpha \leq t_j\}$. In turn, to obtain FCP control at level $\alpha$ with probability $\geq 1 - \delta$ we choose $\widehat{\alpha} = t_{\lfloor \alpha m\rfloor}$. Therefore, the FCP-controlling intervals can be explicitly written $\mathbf{\mathcal{C}(\widehat{\alpha})} = (C_{i, \widehat{\alpha}})_{i \in [[m]]}$ with $C_{i, \widehat{\alpha}} = \left[\hat{\mu}\left(X_{n+i}\right) \pm S_{(\lceil(n+1)(1-\widehat{\alpha})\rceil)}\right]$. The confidence intervals only depend on the threshold family via $\widehat{\alpha}$. We have added this explanation to the updated version of the manuscript.
>
> **Template optimality**: while our experiments unambiguously show that CoJER leads to tighter FCP control than the bounds of Gazin et al, we currently do not have theoretical support for this result. In fact, even the formal definition of an optimal shape is not trivial, and we have left this exciting perspective for future work.
>
> > Does the model aggregation scheme preserve the coverage properties of CP? Does this follow from JER control?
>
> CoJER does not ensure a marginal coverage guarantee for each test observation when $m>1$. Indeed, we argue that this type of guarantee is not interpretable in the transductive case considered here. Instead, CoJER offers a strong probabilistic guarantee (FCP control) over the entire set of $m$ test observations.  However, note that for a single observation ($m=1$), FCP control is equivalent to the marginal coverage guarantee offered by SCP.
>
> **Significance level and comparison to the Benjamini-Hochberg procedure**: indeed, the reviewer's intuition is correct. CoJER builds a JER controlling family $t$ and outputs an adjusted level $\widehat{\alpha} = t_{\lfloor \alpha m\rfloor}$ for which the confidence intervals $C(\widehat{\alpha})$ controls the FCP at level $\alpha$. This is markedly different from the Benjamini-Hochberg (BH) procedure. First, BH outputs a **rejection set** for which there is a statistical guarantee. In the transductive setting of conformal prediction, we are interested in **obtaining a guarantee for all test points simultaneously** and not only for a certain subset. This renders BH and other similar methods unapplicable in this context. Second, BH offers a guarantee on the **expected** proportion of False Discoveries and not in probability.
>
> **Title**: we agree that the title is too vague and doesn't clearly distinguish the contribution. We propose to rename the paper **False Coverage Proportion control for Conformal Prediction**. To this end, we have sent a message to the PCs of the conference.

---

> > ### Comment · Reviewer_ExJ2 · 2025-04-04
> >
> > Thank you for the helpful clarifications. I have raised my score. The authors have addressed my concerns. Overall, I think the paper makes a noteworthy contribution, and the authors' revisions help address my main concern, which was the clarity of writing.

---

> > > ### Author Response · Authors · 2025-04-07
> > >
> > > We thank the reviewer for their prompt response and for raising their score. We remain at the reviewer's disposal should they have any additional questions.

---

### Official Review · Reviewer_FJKa · 2025-03-14

**Overall Recommendation:** 3

**Summary:**

The paper introduces CoJER (Conformal Joint Error Rate), a novel method designed to improve the reliability of split conformal prediction (SCP) by controlling the False Coverage Proportion (FCP). While traditional SCP ensures marginal coverage over multiple test points, it does not guarantee that the proportion of test points outside their confidence intervals remains controlled with high probability. This limitation is crucial in real-world settings, where multiple predictions are made simultaneously, such as in healthcare and finance.

The authors propose CoJER, which leverages conformal p-values and Joint Error Rate (JER) control to obtain tight FCP bounds. Unlike prior work (e.g., Gazin et al., 2024), which uses fully parametric bounds, CoJER provides a nonparametric calibration procedure that ensures sharper confidence intervals. The method is extended to model aggregation, allowing for robustness across different predictive models. Extensive experiments on 17 OpenML datasets demonstrate that CoJER achieves better FCP control while maintaining shorter confidence intervals than existing methods.

**Claims And Evidence:**

The claims made in the paper are generally well-supported by theoretical and empirical evidence. The theoretical guarantees for FCP control are rigorously derived, and the connection between conformal p-values, JER control, and FCP bounds is well-established. The proposed method is thoroughly compared to existing SCP-based approaches, particularly the parametric method of Gazin et al. (2024), which is known to be overly conservative.

Empirically, the extensive evaluation across multiple datasets and models confirms that CoJER:

Achieves valid FCP control while standard SCP fails.
Produces shorter confidence intervals compared to existing FCP-controlling methods.
Remains robust across different model choices through the aggregation framework.
However, one potential limitation is that the paper primarily evaluates performance on tabular regression datasets. The applicability of CoJER to classification problems or high-dimensional deep learning models is not explored in depth.

**Essential References Not Discussed:**

The paper provides a comprehensive review of prior work, but could benefit from including:

Adaptive Conformal Inference: Angelopoulos et al. (2023) propose an adaptive method for controlling conformal prediction widths, which may be relevant for understanding how CoJER adapts to different datasets.
Resampling-based Conformal Methods: Recent work on Jackknife+ (Barber et al., 2021) explores alternatives to split conformal prediction, which could be useful for comparison.

**Experimental Designs Or Analyses:**

The experimental design is robust and methodologically sound:

Multiple datasets: The evaluation on 17 OpenML datasets ensures that results are not dataset-specific.
Multiple models: The experiments consider Random Forest, MLP, SVR, KNN, and Lasso, providing insights into method robustness across different modeling paradigms.
Multiple baselines: The comparison includes standard SCP, the Gazin et al. (2024) method, and CoJER, making it comprehensive.
One area for improvement is the lack of ablation studies on:

The effect of different choices of transformation functions in CoJER.
The impact of varying JER thresholds on FCP control.
A deeper exploration of these factors could provide more insights into when and why CoJER outperforms existing methods.

**Methods And Evaluation Criteria:**

Yes, the methodology and evaluation criteria are well-aligned with the problem. The paper focuses on realistic tabular prediction tasks where ensuring tight and reliable confidence intervals is critical. The choice of OpenML datasets provides a diverse set of benchmarks, making the results more generalizable.

The evaluation metrics—FCP control, interval length, and empirical coverage rates—are appropriate for assessing the effectiveness of the proposed method. However, additional benchmarks on structured datasets (e.g., time-series, NLP tasks) could further strengthen the paper’s claims about general applicability.

**Other Comments Or Suggestions:**

Figure 1 & 2: Labels could be clearer—mentioning "Relative Interval Length" explicitly in the y-axis would help.
Section 5: Define "$$\delta$$" earlier to improve readability.

**Other Strengths And Weaknesses:**

Strengths:
Well-motivated problem: FCP control is a crucial extension to standard SCP.
Mathematically rigorous: The derivations are well-grounded in statistical theory.
Efficient and practical: CoJER remains computationally comparable to SCP.
Weaknesses:
Limited exploration of classification problems.
No ablation studies to test different parameter choices.
Monte Carlo estimation reliance—more discussion on computational trade-offs is needed.

**Questions For Authors:**

How does CoJER perform on classification problems?
What is the computational overhead compared to standard SCP?
Can the method be extended to structured prediction tasks?

**Relation To Broader Scientific Literature:**

This paper builds upon two key strands of research:

Conformal Prediction: The work extends Split Conformal Prediction (Lei et al., 2018; Vovk et al., 2005) to handle False Coverage Proportion (FCP) control.
Multiple Testing & JER Control: The approach adapts Joint Error Rate control (Blanchard et al., 2020) to derive nonparametric FCP bounds.
The primary novelty lies in:

Reformulating FCP control using conformal p-values and JER-based calibration.
Providing tighter, nonparametric bounds compared to prior parametric approaches.
Extending conformal aggregation techniques to improve robustness across models.

**Theoretical Claims:**

The proofs presented in Sections 3 and 4 appear to be mathematically sound. The derivation of JER-based FCP bounds follows from standard techniques in conformal inference and multiple testing. Specifically:

Proposition 1 (characterization of the joint distribution of conformal p-values) is correctly cited from Gazin et al. (2024).
Proposition 2 (link between FCP and JER) follows logically from Lemma 1 and prior work on multiple hypothesis testing.
Proposition 3 & 4 (nonparametric JER control) leverage Monte Carlo approximations, which are empirically validated.
I did not find any major issues in the proofs, but a more detailed discussion on the asymptotic behavior of CoJER for large-scale datasets would be beneficial.

---

> ### Author Rebuttal · Authors · 2025-03-31
>
> We thank the reviewer for their time and insightful comments. Please find our answers to the points raised below.
>
> >However, one potential limitation is that the paper primarily evaluates performance on tabular regression datasets. The applicability of CoJER to classification problems or high-dimensional deep learning models is not explored in depth.
>
> While our experiments focus on tabular regression datasets, we would like to emphasize that **CoJER is fundamentally agnostic to the specific predictive setting** as long as the transductive assumption holds (i.e., access to many test points). This stems from the fact that CoJER operates purely on conformal p-values inherited from the CP framework. As such, the method is applicable to classification tasks and other model families, including high-dimensional deep learning models, provided conformal $p$-values are available.
>
>
> > No ablation studies to test different parameter choices.
>
> The main parameter of CoJER is the aggregation function. We have performed an additional experiment to compare four possible choices: harmonic mean, arithmetic mean, geometric mean and quantile aggregation. In this setting, harmonic mean aggregation outperforms arithmetic mean, geometric mean and quantile aggregation consistently. Please see our answer to reviewer DTau for all experimental details.
>
> > Monte Carlo estimation reliance—more discussion on computational trade-offs is needed. How does CoJER perform on classification problems?
>
> In our setting, we see the reliance on Monte Carlo estimation as a strength rather than a weakness, since it allows JER control with arbitrary precision at a small computation cost. Sampling from $P_{n,m}$ is done in $O(n + m)$. Moreover, this is done once and for all for given values of $n$ and $m$.
>
> > What is the computational overhead compared to standard SCP?
>
> In the transductive setting considered in this paper with $m$ test points, both SCP and CoJER require $O(m n \log(n))$ to obtain $m$ conformal $p$-values (each of them requires sorting $n$ conformity scores). For FCP control with $B$ MC samples, CoJER additionally requires $O(B m \log(m))$ for template generation and sorting, and $O(B m (\log(m) + \log(B)))$ for calibration using binary search.
>
> Therefore, neglecting the logarithmic terms for simplicity, the complexity of standard SCP is $O(m n)$ and that of CoJER is $O(m (n+B))$, where $B$ is the number of MC samples (which is user-defined does not depend on $n$ or $m$). In particular, the complexities are of the same order if $B$ is chosen to be of the same order as $n$.
>
> > The paper provides a comprehensive review of prior work, but could benefit from including: Adaptive Conformal Inference: Angelopoulos et al. (2023) propose an adaptive method for controlling conformal prediction widths, which may be relevant for understanding how CoJER adapts to different datasets. Resampling-based Conformal Methods: Recent work on Jackknife+ (Barber et al., 2021) explores alternatives to split conformal prediction, which could be useful for comparison.
>
> Both of these approaches provide marginal risk control, i.e. *for a single test point*. As argued above in our reply to reviewer 8Xty, such approaches could be leveraged to control the False Coverage Rate (FCR) but they do not provide FCP control in probability. Therefore, they are not more relevant than SCP as competitors for our method.

---

> > ### Comment · Reviewer_FJKa · 2025-04-07
> >
> > I thank authors for addressing my concerns. I have raised my score.

---

> > > ### Author Response · Authors · 2025-04-07
> > >
> > > We thank the reviewer for their prompt response and their decision to raise their score. However, unless we are mistaken, the score remains unchanged. We would be grateful if the reviewer could confirm whether the update was submitted in case there is a technical issue.

---

### Official Review · Reviewer_8Xty · 2025-03-14

**Overall Recommendation:** 4

**Summary:**

This paper examines the limitations of Split Conformal Prediction (SCP) in controlling the False Coverage Proportion (FCP) across multiple predictions. While SCP ensures control over the False Coverage Rate (FCR), it does not provide high-probability guarantees on the actual proportion of non-covered intervals across multiple test points.  To address this gap, the authors propose CoJER, a method specifically designed for FCP control in multi-point prediction settings. By reformulating SCP as a p-value thresholding procedure, they derive conformal p-values within a Joint Error Rate (JER) control framework, avoiding strong parametric assumptions. This results in a more adaptive and sharper bound compared to the prior work of Gazin et al. (2024).   Additionally, the authors extend their method to aggregate conformal prediction intervals across different models, enhancing robustness and reducing sensitivity to specific modeling choices. Their results demonstrate that CoJER effectively controls FCP under any aggregation scheme*, offering a principled approach to multi-point uncertainty quantification.

**Claims And Evidence:**

The claim that "CoJER yields shorter intervals than the state-of-the-art method for FCP control and only slightly larger intervals than standard SCP." is empirically validated using benchmark datasets (OpenML).

However, the evaluation setup raises concerns. Since SCP does not ensure FCP control, it should not be included in interval length comparisons, as its intervals may be shorter at the expense of failing to meet the desired coverage guarantees. Consequently, the evaluation effectively compares CoJER against a single state-of-the-art method, limiting the strength of the claim. A more comprehensive comparison, potentially including additional baseline methods or alternative strategies for FCP control, would strengthen the evidence supporting this statement.

**Essential References Not Discussed:**

The only two works essential for understanding this paper are Gazin et al. [3] and the research on joint error rate [4], both of which are extensively discussed.

**Experimental Designs Or Analyses:**

I verified the validity of the experimental design. They used 17 OpenML datasets, performed 30 dataset splits, and tested five different regression models, providing a reasonable basis for confidence in the results. The choice of $ \alpha = \delta = 0.1 $ is appropriate based on the literature.

**Methods And Evaluation Criteria:**

Using 17 OpenML datasets provides a solid foundation for evaluation. Additionally, reporting the relative interval length compared to the shortest interval across all methods, averaged over multiple splits per dataset, offers a clear basis for comparison. However, incorporating domain-specific real-world datasets from fields such as medicine or finance would have further strengthened the study, given the stated relevance of multi-point prediction in these areas.

**Other Comments Or Suggestions:**

The title of the paper is too broad, as achieving tight and reliable confidence intervals is a fundamental goal of any conformal prediction method. A more precise title should reference *CoJER* and its role in *FCP control* to better reflect the paper's specific contribution.

### *Questions:*
- *Line 62 (Page 2):* How do you define *"prediction intervals with a size close to the optimal length"*? What constitutes the optimal length in this context?
- What is the impact of $\delta $ on performance? Could setting $\delta $ to a very low value be beneficial?

### *Minor Suggestions:*
- Could you provide relevant examples of multiple test point settings?
- Most equations lack numbering. While this improves clarity, adding numbers would enhance readability and referencing.
- *Line 225:* The reference to *"Appendix B"* is not clickable.
- The legends in *Figures 1 and 2* would be more readable if placed above the plots.

**Other Strengths And Weaknesses:**

### *Strengths:*
- Solid mathematical foundation with a well-structured flow of ideas.
- Strong empirical results that clearly demonstrate the superiority of the proposed method.

### *Weaknesses:*
- The title is too broad; *"Tight and reliable conformal prediction"* is a common goal in this field and does not clearly distinguish the contribution.
- Lacks sufficient intuitive explanations for key steps, with some missing motivations (e.g., the choice of the harmonic mean for the aggregation scheme).
- Limited comparison, as it considers only two methods, one of which does not ensure FCP control, restricting the strength of the empirical evaluation.

**Questions For Authors:**

See above.

**Relation To Broader Scientific Literature:**

The paper lacks a dedicated related work section, which would help contextualize previous methods and enhance understanding. While the connection to SCP and the approach by Gazin et al. is well-established, a broader comparison with other risk-control methodologies should have been explored—particularly *conformal risk control* [1] and *risk-controlling prediction sets* [2]. Integrating these perspectives would provide a more comprehensive view of how CoJER fits within the broader landscape of uncertainty quantification and risk control.


[1] Angelopoulos, Anastasios N., Stephen Bates, Adam Fisch, Lihua Lei, and Tal Schuster. 2022. “Conformal Risk Control.”
[2] Bates, Stephen, Anastasios Angelopoulos, Lihua Lei, Jitendra Malik, and Michael I. Jordan. 2021. “Distribution-Free, Risk-Controlling Prediction Sets.”

**Theoretical Claims:**

The proofs for Lemma 2, Proposition 2, and Proposition 3 appear sound.

---

> ### Author Rebuttal · Authors · 2025-03-31
>
> We thank the reviewer for their time and insightful comments. Please find our answers to the points raised below.
>
> >However, the evaluation setup raises concerns. Since SCP does not ensure FCP control, it should not be included in interval length comparisons [...] including additional baseline methods or alternative strategies for FCP control, would strengthen the evidence supporting this statement.
>
> To the best of our knowledge, the only existing approach that explicitly targets FCP control in CP is the work of Gazin et al. We would be glad to include additional baselines if others are brought to our attention, and we welcome any suggestions in that regard.
>
> We agree that a fair comparison of interval lengths should ideally be restricted to methods that control the false coverage proportion (FCP). We chose to include SCP, despite it not controlling the FCP, to illustrate that CoJER provides substantially stronger coverage guarantees with only a modest increase in interval length.
>
> > The paper lacks a dedicated related work section [...]  other risk-control methodologies should have been explored—particularly conformal risk control [1] and risk-controlling prediction sets [2]."
>
> For a new test point $(X_{n+1},Y_{n+1})$, SCP provides a confidence interval $C_{\alpha}\left(X_{n+1}\right)$ with the following guarantee:
> $\mathbb{P}\left[Y_{n+1} \notin C_{\alpha}\left(X_{n+1}\right)\right] \leq \alpha$.
> The two approaches mentionned by the referee extend SCP as follows:
> - Conformal risk control replaces marginal miscoverage by a general loss $\ell$ , aiming for the guarantee: $\mathbb{E} \left[\ell(C(X_{n+1}, Y_{n+1})\right] \leq \alpha$
> - The approach in [2] provide the same guarantees as SCP, in the case of set-valued prediction.
>
> As such, these approaches still provide *marginal risk control for a single test point*. Our paper focuses on the transductive setting, where $m$ such test points are available. In this setting, while the marginal guarantees could be leveraged to control the False Coverage Rate (FCR), they do not provide FCP contol in probability. Therefore, they are not more relevant than SCP as competitors for our method. We have added text to the section 2.1 "Split conformal prediction" of the manuscript to clarify this important point.
>
>
> > Line 62 (Page 2): How do you define "prediction intervals with a size close to the optimal length"? What constitutes the optimal length in this context?
>
> In this context, we meant that CoJER produces prediction intervals with lengths close to those of SCP, while additionally providing formal FCP control—something SCP does not guarantee. We agree that the term "optimal" could be misleading and have reformulated this point in the updated version of the paper to avoid ambiguity.
>
> >What is the impact of $\delta$ on performance? Could setting  to a very low value be beneficial?
>
> FCP is controlled with probability greater than $1 - \delta$. As $\delta$ becomes smaller, FCP control becomes increasingly stringent. In terms of performance, this means that decreasing $\delta$ increases interval width. Setting $\delta$ to a very low value (e.g. $\delta = 0.001$) would ultimately lead to a non-informative statement, with FCP control holding with overwhelming probability, for very wide intervals.
>
> >The title is too broad; "Tight and reliable conformal prediction" is a common goal in this field and does not clearly distinguish the contribution.
>
> We agree that the title is too vague and doesn't clearly distinguish the contribution. We propose to rename the paper **False Coverage Proportion control for Conformal Prediction**. To this end, we have sent a message to the PCs of the conference.
>
> >Lacks sufficient intuitive explanations for key steps, with some missing motivations (e.g., the choice of the harmonic mean for the aggregation scheme).
>
> Regarding the choice of the harmonic mean, we have performed an additional experiment to compare four possible choices: harmonic mean, arithmetic mean, geometric mean and quantile aggregation. Please see our answer to reviewer DTau for all experimental details.
>
> >Limited comparison, as it considers only two methods, one of which does not ensure FCP control, restricting the strength of the empirical evaluation.
>
> Please see our answer above: to the best of our knowledge, the work of Gazin et al. is the only existing approach that explicitly targets FCP control in CP. We would be glad to include additional baselines if others are brought to our attention, and we welcome any suggestions in that regard.
>
> > Could you provide relevant examples of multiple test point settings?
>
> A common example of multiple test point setting in CP are real-time systems with batches: In applications like fraud detection or content recommendation, models may process incoming data in batches for efficiency. Another common setup is the offline evaluation of predictive models: before deploying a model, it is often evaluated on a fixed test set.

---

> > ### Comment · Reviewer_8Xty · 2025-04-06
> >
> > We thank the authors for their response. I have raised my score.

---

> > > ### Author Response · Authors · 2025-04-07
> > >
> > > We thank the reviewer for their prompt response and for raising their score. We remain at the reviewer's disposal should they have any additional questions.

---

### Official Review · Reviewer_DTau · 2025-03-14

**Overall Recommendation:** 3

**Summary:**

This paper addresses the challenge of controlling the False Coverage Proportion (FCP) in split conformal prediction (SCP). While SCP provides computationally efficient confidence intervals, it only guarantees marginal coverage over multiple test points. The authors highlight that in real-world scenarios, where multiple predictions are made simultaneously, the FCP of standard conformal prediction algorithms fluctuates significantly. This work proposes CoJER, a novel Joint Error Rate (JER) control-based method that achieves tight and reliable FCP control using a refined characterization of conformal p-values in a transductive setting. The authors also extend this procedure to provide FCP control under any pre-specified aggregation scheme for using knowledge from multiple prediction models simultaneously.

**Claims And Evidence:**

The claims made in the paper are supported by clear and convincing evidence.

**Essential References Not Discussed:**

The authors have provided a detailed overview of all relevant related works.

**Experimental Designs Or Analyses:**

The experimental design and analysis is sound.

**Methods And Evaluation Criteria:**

The datasets, baselines, and evaluation metrics considered for the problem are appropriate.

**Other Comments Or Suggestions:**

N/A

**Other Strengths And Weaknesses:**

Strengths
- The paper addresses an important issue.
- The JER procedure and the aggregation procedure provide strong theoretical guarantees without any strong assumptions beyond the standard exchangeability assumption.
- The experimental evidence shows a clear advantage of using the proposed algorithms over the existing state-of-the-art.

Weaknesses
- The paper is very dense and hard to read. Some toy examples / walk-throughs of the procedure could be quite useful for understanding.
- The template-building procedure is not described well.
- The matrix dimensions are mentioned to be n x p, but p is never mentioned before. This is quite confusing. (It might be a typo making, and it should be n x m)
- The authors don't theoretically or experimentally explore the relative tightness of the finite sample FCP bounds depending on aggregation mechanism used.

**Questions For Authors:**

Please refer to the Weaknesses section above.

**Relation To Broader Scientific Literature:**

This work provides a way to control the FCP over a given set of test points with high probability. It also extends the guarantees to aggregate the knowledge from multiple models to provide more efficient conformal intervals. While there has been previous work on both of these topics, this work offers a new perspective by using the p-value interpretation of conformal intervals to better use the rich literature on Joint Error Control using p-values. The algorithms mentioned in the paper improve the applicability of the conformal prediction framework to more real-world scenarios.

**Theoretical Claims:**

The proofs of the theoretical claims are accurate.

---

> ### Author Rebuttal · Authors · 2025-03-31
>
> We thank the reviewer for their time and insightful comments. Please find our answers to the points raised below.
>
> > The paper is very dense and hard to read. Some toy examples / walk-throughs of the procedure could be quite useful for understanding.The template-building procedure is not described well.
>
> To improve the clarity of the paper, we will add a pedagogical figure that explains the procedure in two steps in the updated manuscript. The first panel illustrates the concept of JER control with an example of template and of calibrated threshold family. The second panel illustrates the intervals obtained with CoJER using the adjusted risk level $\widehat{\alpha} = t_{\lfloor \alpha m\rfloor}$ with $t$ the calibrated threshold family. We will also add pedagogical interpretations for the notions of threshold families and templates.
>
> >The matrix dimensions are mentioned to be n x p, but p is never mentioned before. This is quite confusing. (It might be a typo making, and it should be n x m)
>
> Thanks for pointing this out, this is indeed a typo -- the matrix dimensions are $n \times m$. We have corrected this in the updated version of the manuscript.
>
> >The authors don't theoretically or experimentally explore the relative tightness of the finite sample FCP bounds depending on aggregation mechanism used.
>
> To address the reviewer's concern, we performed an additional experiment on the 17 openML datasets used in the paper. In this experiment, we compare four possible aggregation schemes: harmonic mean, arithmetic mean, geometric mean and quantile aggregation [1].
>
> We use the setup described in the main text - i.e. $\alpha = 0.1, \delta = 0.1$. Importantly, we first check that the FCP is controlled for all types of aggregation by reporting the FCP event non-coverage as described in the main text. We also compute the (relative) interval width for each aggregation scheme, averaged across 20 splits.
>
> |                                   | Harmonic mean | Geometric mean | Arithmetic mean | Quantile aggregation |
> |-----------------------------------|---------------|----------------|-----------------|----------------------|
> | FCP event coverage                | 94%           | 100%           | 100%            | 100%                 |
> | Interval width increase (vs best) | **0%**            | +24%           | +230%           | +54%                 |
>
>
> Coherently with the theoretical guarantees obtained in Proposition 4, **the FCP is controlled for all four aggregation schemes.** In terms of interval tightness, harmonic mean aggregation outperforms arithmetic mean, geometric mean and quantile aggregation consistently.
>
> ### References
> [1] Meinshausen, N., Meier, L., & Bühlmann, P. (2009). P-values for high-dimensional regression. Journal of the American Statistical Association, 104(488), 1671-1681.

---

### Decision · Program_Chairs · 2025-05-01

**Decision:**

Accept (poster)

**Comment:**

The overall recommendations are: 3, 4, 3, 3

Paper summary: this paper proposes a method CoJER to control FCP (false coverage portion) by controlling JER (joint error rate), so that it concentrates around FCR (false coverage rate). The underlying JER controlling framework is to characterize the distribution of p-values in a transductive setting.

Reviewers agree that this paper considers an important and well-motivated problem. The proposed method has strong theoretical and empirical results. Every reviewer gives positive ratings to support accepting this paper.